# Parental experience of change following VIPP-SD in families with preschool children with externalizing behavior: A qualitative interview study

**Nina M. Lassen** [1]\*, **Tine Steenhoff**[1], **Bryan Cleal**[2], **Amanda Frees**[1], **Mette S. Væver**[1]

**1** Department of Psychology, Centre of Excellence in Early Intervention and Family Studies, University of Copenhagen, Copenhagen, Denmark, **2** Copenhagen University Hospital, Steno Diabetes Center Copenhagen, Copenhagen, Denmark

\* nina.marie.lassen@psy.ku.dk

## Abstract

### Background

Insensitive parenting and ineffective disciplinary strategies are known risk factors for child externalizing behavior. The Video-feedback Intervention to promote Positive Parenting and Sensitive Discipline (VIPP-SD) has documented effect in promoting sensitive parenting, but little is known on how VIPP-SD is experienced by parents. This study explores how parents of preschool children with externalizing behaviors experience change following VIPP-SD delivered by trained childcare providers.

### Methods

Individual qualitative semi-structured interviews were conducted with 9 mothers and 2 fathers to explore the parents' experiences of change following the intervention. Data were analyzed using Braun and Clarke's reflexive thematic analysis.

### Results

Four themes were generated: 1) "All of her behavior is actually just a result of how she feels, right?"—Enhanced parental understanding, 2) Meeting the child's needs in comfort and in play, 3) Learning to prevent and manage conflicts is essential–diverse experiences of gains and progress, 4) "I'm actually not a bad parent"–new positive perspectives.

### Conclusion

Parents experienced an enhanced capacity to understand their child and positive development in their parenting behavior, skills and confidence as well as improvements in the parent-child relationship after receiving VIPP-SD. Findings also suggest potential areas for adaptation of VIPP-SD when intervening in families with a child exhibiting externalizing behaviors, as parental experiences of gains related to conflict management varied. Further research on this matter is recommended.

**Data Availability Statement:** Data cannot be shared publicly because our dataset involves subjects who did not explicitly consent to the public access of their pseudonymized data, which

is required to comply with the Danish Data Protection Act. Therefore, publishing the dataset does not align with the ethical standards of data management and regulations upheld at the University of Copenhagen. The interviews may also contain sensitive information and cannot be fully de-identified. Upon request, anonymized extracts and original parts of the data can be obtained by contacting the Data Management and Information Security department at the Department of Psychology, University of Copenhagen (contact via legal@samf.ku.dk), for researchers who meet the criteria for access to confidential data.

**Funding:** The study was funded by the Independent Research Fund Denmark. The funders had no role in study design, data collection and analysis, decision to publish, or preparation of the manuscript.

**Competing interests:** NML and TS are certified VIPP-SD interveners and are currently training to become VIPP-SD trainers. They anticipate undertaking paid supervision and training in the future. All other authors declare no known conflicts of interest.

# Introduction

Externalizing behaviors, such as non-compliance, aggression, hyperactivity and oppositional behaviors are frequently observed among young children, and the onset often occurs during preschool years [1,2]. A general population study, including reports from more than 6000 caregivers, found a prevalence of 6.5% for moderate externalizing problems and emotionally reactive behavior among preschool children [3]. For many young children, externalizing behavior is a normative and transitional phenomenon [3–5]. However, persistent and severe externalizing behavior problems in childhood is associated with a range of later negative outcomes, such as antisocial problem behaviors, emotional dysregulation, academic underachievement, peer victimization and rejection, risk of psychiatric problems and negative long-term economic outcomes [5–11]. Harsh and insensitive parenting is a known risk factor for the development of externalizing behaviors problems [10,12–17]. Thus, promoting sensitive parenting through interventions is a key strategy for preventing persistent externalizing problems. Video-feedback Intervention to promote Positive Parenting and Sensitive Discipline [18] is a parenting intervention that aims to enhance sensitive parenting to support secure attachment relationships, while also promoting sensitive disciplinary strategies to reduce child externalizing behavior [19]. A recent meta-analysis of 25 randomized controlled trials (RCTs) has established evidence for the effectiveness of VIPP-SD on caregivers' sensitivity, sensitive disciplining behavior, child attachment security, but not on reduction of child externalizing behavior problems [20]. However, a RCT ($N = 300$) found VIPP-SD to be effective in reducing behavior problems in children aged 12–36 months [21]. To date, only a few studies have qualitatively explored how parents of children with externalizing problems experience modified versions of VIPP-SD. Qualitative evaluations of outcomes, which focus on participants' understandings of *what has changed*, can complement the predominant tradition of quantitative outcome assessment, and provide insights into possible changes and pathways for change not otherwise captured [22]. Parental reports of experienced change can illuminate the perceived impact and helpfulness–as well as the absence of expected change–of VIPP-SD.

## Video-feedback intervention to promote positive parenting and sensitive discipline

A number of parenting programs aim to promote parental sensitivity and/or efficient disciplinary strategies, as low parental sensitivity and inefficient disciplinary behavior have been found to predict negative socio-emotional development in the child [23–26]. One such intervention is VIPP-SD, where the original version focused on promoting parental sensitivity and secure attachment, and an additional focus on sensitive discipline (SD) was added later [19,27]. The VIPP-SD intervention aims to provide parents with strategies for managing difficult child behaviors in addition to promoting sensitive parenting, thereby addressing an overarching goal of reducing child behavior problems and preventing further development of antisocial behavior [19]. Informed by Ainsworth's concept of parental sensitivity, VIPP-SD focuses on promoting parents' observational skills and awareness of the child's signals, the parental ability to correctly interpret these signals and to respond to them adequately and promptly [28]. Sensitive parenting is addressed through four themes that constitute the structure and progression of the intervention (see Table 1). The sensitive discipline component is addressed in parallel through four discipline themes (see Table 1) and is based on an integration of attachment theory and Patterson's [29] coercion theory, supplemented with Hoffman's [30] constructs on inductive discipline and empathy [19,27]. Coercion theory originates from social learning theory, where reinforcement processes are considered an important factor for overall personality development [19]. Effective disciplining is, among other, suggested to be supported by parents reinforcing

**Table 1. Structure and themes of the VIPP-SD intervention for each session.**

| Session | Sensitive Parenting | | Sensitive Discipline | |
|---|---|---|---|---|
| | Theme | Focus | Theme | Focus |
| 1. | Introduction and filming (no feedback) | | Introduction and filming (no feedback) | |
| 2. | 1. Exploration versus attachment behavior | Supporting the parent's observational skills by helping the parent distinguish between the child's exploratory behavior and attachment behavior. | 1. Distraction and Induction | Encourage using distraction and induction in response to challenging child behavior or conflict-evoking situations. Induction (explaining) as a strategy to help the child internalize parental rules and develop empathy. Distraction (e.g. suggestions of alternatives or postponing attractive activities) as a strategy to support child compliance. |
| 3. | 2. Speaking for the child | Encouraging observational skills and accurate interpretation of the child's signals by verbalizing both facial expressions and non-verbal cues. | 2. Positive reinforcement | Encourage the use of positive reinforcement e.g. praising and acknowledging the child for positive behavior and not giving attention to challenging behavior/negative attention seeking. |
| 4. | 3. Sensitivity chain | Illustrate the importance of the prompt and adequate parental response to the child's signals by showcasing positive interaction moments ("sensitivity chains") consisting of three steps; the child's signal, the parent´s response and the child's reaction to the parent's response. | 3. Sensitive pause | Focus on how to use the "sensitive pause" to deescalate heated conflicts sensitively. |
| 5. | 4. Sharing emotions | Encourage and highlight moments of shared emotions and parental attunement to both negative and positive feelings within the child. | 4. Empathy for the child and induction | Encourage the parent in showing empathy for the child, while consistently making use of the discipline strategies and clear limit-setting. |
| 6. | Booster, co-parent are invited to join | Repetition and reinforcement | Booster, co-parent is invited to join | Repetition and reinforcement |
| 7. | Booster, co-parent are invited to join | Repetition and reinforcement | Booster, co-parent is invited to join | Repetition and reinforcement |

[18–20,27,31].

positive child behavior and ignoring negative behavior. In VIPP-SD, this approach is supplemented by additional non-coercive strategies, such as distraction and induction. Coercion theory and attachment theory derive from different theoretical traditions, but both emphasize the significance of contingent and non-aversive parent-child interactions [19,27].

VIPP-SD is a brief home-based intervention targeting 1–6-year-old children and their parents. It includes seven sessions of 1½-2 hours each with a 2-4-week interval. The use of video-feedback is central and is hypothesized to be an active intervention component [20]. The targeted parent and child are videotaped during daily interactions, such as playing together, clean-up situations and mealtimes. The VIPP-SD intervener analyzes the video and prepares feedback for the parent according to the VIPP-SD manual [18]. In the feedback, the intervener reinforces positive parent-child interactions and appropriate disciplinary strategies. The intervention is both standardized, following a specific thematic progression through the use of a manual (see Table 1) and personalized through the use of individualized video material, analysis and feedback, hereby allowing for implementation in various contexts [20].

## Previous qualitative research on VIPP-SD

To the best of our knowledge, parents' and other caregivers' experience of VIPP-SD has only been examined qualitatively in relation to adapted versions of VIPP-SD. Specifically, six feasibility/pilot studies have qualitatively explored parents' and other caregivers' experiences. Using interviews, Barnicot and colleagues [32] reported on 23 mothers' experiences of VIPP-PMH (VIPP-SD adapted for a perinatal mental health context with focus on perinatal

'personality disorder') and Williams [33] analyzed interviews on 14 fathers' experience, where two received VIPP-SD individually and 12 received an adapted version of VIPP-SD with their partner. Using a mixed method design, Dugmore and colleagues [34] explored eight adoptive parents' experience with VIPP-Family Placement, Oliveira and colleagues [35] explored 11 foster carers' experiences of VIPP-Foster Care, and Iles and colleagues [36] explored five families' experience with receiving VIPP-Co, designed for parents together. Further, Starreveld and colleagues [37] explored three primary school teachers' experience with VIPP-School. Across all six studies, participants reported gaining a better understanding of their child and their child's signals [32–37]. This included a more nuanced and positive interpretation of the child's behavior and emotions. Additionally, the studies reported that parents/caregivers experienced positive change in their interactions, specifically in how they responded and adapted to their child's signals and needs [32–37]. Several of the studies furthermore reported experiences of improvements in the parent/caregiver-child relationship [33–35]. Another recurring finding was that participants felt reassured and more confident about their parenting/caregiving following the intervention [32–36]. Some studies found that participants reported acquiring helpful strategies to manage difficult child behaviors [32,33,35], or developing a different and more positive perspective on challenging child behaviors [32,37]. However, in the study by Williams [33], some fathers experienced dissatisfaction with the level of feedback received regarding their management of specific challenges and reported no increase in confidence in managing difficult behaviors following the intervention. In the study by Iles and colleagues [36] a theme reflects the parents' difficulties in generalizing learnings from the VIPP-Co intervention to daily situations, specifically regarding challenging child behaviors. Overall, the studies find predominantly positive experiences from participating in various adaptations of VIPP-SD. However, they also indicate challenges in meeting the needs of families with children exhibiting externalizing behaviors, warranting further exploration of this specific group. Furthermore, to date, no qualitative study has targeted parents of preschool children (aged 3–5 years) exhibiting externalizing behaviors and explored their experiences of outcomes of VIPP-SD.

### The current study

The present study is, to the best of our knowledge, the first to focus exclusively on the experience of *change* among parents of preschool children with externalizing behaviors following the VIPP-SD-intervention. We focused on exploring changes that the parents report regarding themselves, their child, and experienced changes in parenting and the parent-child relationship. The study thus has the potential to provide new insights on perceived outcomes of the intervention for this group, contributing to the existing knowledge on VIPP-SD as a facilitator of change in families struggling with externalizing problems.

## Methods

This study was approved by The Institutional Ethical Review Board, University of Copenhagen, Department of Psychology (approval number IP-IRB/14122020). All adult participants gave written consent to participate in the study and parental consent was collected for all participating children as well. The names of the participants in connection with presented extracts have been anonymized.

### Study setting

The current study is a part of a feasibility study of VIPP-SD conducted in two Danish municipalities, where VIPP-SD-trained childcare providers (pedagogues and specialized pedagogical professionals) delivered VIPP-SD to parents of children with externalizing behavior.

## Recruitment

Families could participate in the study if their child was at risk for developing externalizing behavior problems by showing extended amounts of aggressive, non-compliant and/or hyperactive behavior in the childcare center and/or at home. A list with descriptions of externalizing behaviors and examples of how these may be expressed by the child were provided for parents and professionals to identify relevant families. The list entailed 12 examples of externalizing behaviors with inspiration from the SDQ-questionnaire and items used to screen for behavioral- and attention problems, such as "often fights with other children or bullies them", "often loses temper" and "restless, overactive, cannot stay still for long". Inclusion criteria for the study were that the child's behavior should match at least two of the 12 descriptions, and parents and/or professionals were concerned for the child's socio-emotional development. Childcare providers, childcare managers and public health visitors could refer families to the study. Information was also posted on the childcare centers' online communication platform, so that parents who recognized their child's behavior in the description, could sign up for the study themselves. Families were excluded if the child had other problems than externalizing behavior, e.g., internalizing behavior problems, known developmental disorders (e.g. diagnosis of autism), or if there was known parental substance abuse, domestic violence or child maltreatment. Recruitment period for this study: April 1.st 2021 –July 1.st 2022.

## Participants

15 families were included and received the VIPP-SD intervention. For the current study, interview data from 11 families were included. Three families were excluded as they had a VIPP-SD intervener who was a familiar childcare provider from their child's childcare center, which made the parental experience different from the rest, and one family was excluded due to substantial deviations from the standard VIPP-SD procedure by the intervener. Of the 11 families, eight were invited to participate by staff in their child's childcare center, two were invited by their public health visitor, and one mother signed up for the study herself. The participants were nine mothers, and two fathers aged 27–47 years ($M = 36.7$), with eight boys and three girls aged 3–5 years ($M = 3.9$). Overall, the families were well-resourced: all parents were married and living together with the other parent, most had completed either a bachelor- or a master's degree, and none of the participants were unemployed.

## Intervention

When families were included in the study, the target parent would be matched with the first available VIPP-SD intervener. The parents were advised that the target parent should be the one experiencing most challenges with the child and/or the parent who spent most time with the child. The 11 families all received a full VIPP-SD intervention in their homes, delivered by a VIPP-SD-trained childcare provider from one of the two municipalities. The childcare providers either worked in a different childcare center, than the one the family were affiliated with, or were linked to the childcare center as pedagogical consultants. All childcare providers attended a four-day accredited training in VIPP-SD and conducted a complete practice intervention under supervision before starting project trajectories. To ensure the quality of the intervention and adherence to the VIPP-SD manual, each childcare provider further received 3 hours of online supervision per family, which was provided by a certified VIPP-SD supervisor.

We made one adaptation of the original manual to the Danish cultural context: We renamed the concept of "sensitive time-out" to "sensitive pause". The sensitive pause was described to the parents as not just being a pause for the child, but also as a tool and a pause

for parents to manage their own difficult emotions. The change was approved by the VIPP-center at Leiden University and by the original developers of the intervention.

## Data collection

The interviews were conducted by two trained psychology students, within a month after the parents had completed the VIPP-SD intervention. The interviewers were not familiar with VIPP-SD, allowing for authentic curiosity and non-directive questioning. The interviews followed an adapted version of Elliott's Client Change Interview Schedule [38,39]. The CCI explores three main areas in relation to change; the perceived changes by the participants during and after the intervention period, how the participants understand the sources of these changes, including helpful aspects of the intervention, and possible difficult, hindering or missing aspects of the intervention for change [22,39]. The length of the interviews ranged from 57 minutes to 1 hour and 23 minutes. All interviews were audio recorded and transcribed by the two interviewers and a team of research assistants. Transcription were conducted following Clarke and Braun's [40] notation system, inspired by Jefferson [41]. All transcripts were anonymized and QSR International Nvivo 14 software was used as transcription tool.

## Data analysis and reflexivity

Interviews were analyzed using the reflexive thematic analysis (TA) developed by Braun and Clarke [42,43]. TA is a method for developing, analyzing and interpreting patterns of meaning across a qualitative dataset. *Reflexive* TA acknowledges and values the subjective and situated researcher and requires a reflexive approach regarding the researcher's role, research practice and process [44]. TA is best described as an iterative process consisting of six recursive phases: (1) familiarization, (2) coding, (3) generating initial themes, (4) developing and reviewing themes, (5) refining, defining and naming themes, and (6) writing the rapport. Thus, the analysis process moves continuously between data, coding and theme generation. The flexibility of TA made this method suitable for our analysis, both for its capacity to facilitate a combined inductive and deductive theme generation, and in terms of how the method's theoretical flexibility permitted the analysis in later stages to be informed by the theories inherent in VIPP-SD. The coding and main analysis were conducted by (NML) and (TS), both female psychologists and trained VIPP-SD interveners and supervisors. The analysis was further discussed with (BC), (AF) and (MSV), all of whom had no experience with delivering the VIPP-SD. QSR International Nvivo 14 software was used to conduct the coding and initial steps of clustering and analysis. In the familiarization and coding phase, both coders read all transcripts and independently coded three identical interviews to discuss overall impressions of data and to reflect upon similarities and differences in perspectives. This part of the process assisted us in refining our initial understandings of the material and increased our reflexivity by raising our awareness of possible preconceptions and subjective assumptions, such as being particularly attuned to specific areas of change at the expense of others. This was followed by a process of individual coding of respectively five (TS) and six (NML) interviews, before going on to the phases of clustering codes into themes, discussing interconnections and generating thematic maps and final themes in collaboration. The different elements in the analysis (i.e., codes, themes, etc.) were discussed with the research team during each phase, to enhance the plausibility of the interpretation and to serve as a means to reflect on subjectivity. The latter was also supported by the first author keeping a self-reflective journal. In our analysis of the data, we applied a combined inductive and deductive approach. As both coders were researchers with specific interest in early intervention and attachment, as well as being trained VIPP-SD interveners and supervisors, this knowledge by default served as a lens for us to read and code the data, as

well as development of themes [44]. However, the codes and themes were developed through an inductive data-driven approach and were not pre-determined. During the coding process, we continuously reflected on how our positions, values and experience affected our reading of the data and sought to allow room for both the unanticipated, and for possible contradictions, to promote a nuanced interpretation. We coded primarily on the semantic, manifest level. Some code labels however also reflected a more latent, analytical level, when it made sense to explore meanings at a more implicit level, e.g., code labels reflecting expressions of implicitly expressed change/absence of change [44]. Since we wanted to investigate the parents' own understandings, perspectives, and subjective experiences, the epistemological position adopted in this study most closely corresponds to an experiential and critical realist approach. This approach acknowledges the participants' statements as an important source of knowledge, while also acknowledging that perceptions of reality and truth are situated in and shaped by both context and knowledge generation processes [44].

## Results

Our analysis of parents' experience of change following the VIPP-SD intervention resulted in four overall themes presented below and in Table 2. The themes are interrelated, and data extracts often reflect several themes.

### Theme 1: "All of her behavior is actually just a result of how she feels, right?"—Enhanced parental understanding

Theme 1 reflects a salient pattern centering on the parents' experiences of having acquired an enhanced understanding of their children. This change was articulated by parents as a significant benefit from receiving VIPP-SD. Furthermore, it was embedded within their narratives about new ways of interpreting their children's behavior, where they tried to "look behind the behavior" (P1). The parents' descriptions of an improved understanding implied reflections on the child's internal states, intentions, and needs in diverse situations, and how this influenced the child's behavior. For example, in play situations, parents reported that they were now able to interpret child behaviors, such as uneasiness or unfocused behavior, in a more positive and understanding manner. They also experienced this to be the case in everyday life situations, where parents reported an enhanced understanding of the child's reactions to demands, such as the challenges children face in transitioning between activities: "Because naturally, when you take children out of a game or whatever they're interested in, something they're engaged with, well, then—then it's not great, is it, to be taken out of it, right?"(P2)

Some parents described the improved understanding as being helpful in relation to their concerns about more pervasive difficult behavior. One parent, struggling with the challenges of her child's externalizing behavior after becoming an older sister, expressed how she had gained new insights into the reasons behind her child's behavior:

> "She has never, like, done bad things just to be bad, you know. And I've always known that. But I feel like I've had that confirmed and nuanced in my conversations with [the intervener], you know. So, all of her behavior is just a result of how she feels, right? So, when she struggled with a new one coming into the family, you know, um, we've really been focused on her a lot [before the new baby], you know. Um, and we haven't been able to do that the same way, so I think that's why it's been so hard for her." (P3)

Some of the statements revealed how parents now reflected on the contradictions between their *own* wishes and needs and the needs of the child, giving way to a more flexible attitude.

**Table 2. Thematic analysis: Theme overview and descriptions.**

| Themes | Theme | Summary | Illustrative extracts |
|---|---|---|---|
| 1. | "All of her behavior is actually just a result of how she feels, right?"— Enhanced parental understanding | Theme 1 covers the parents' experiences of change in relation to having acquired "a better understanding" of their children as a key change. This is evident both in generalized accounts but also in descriptions of specific situations reflecting an experienced improved ability to reflect on the child's internal states. The theme also encompasses experiences of having gained a better understanding through an achievement of new knowledge regarding typical child development. | "So, my child has had some challenges with aggression, tantrums, and the things we've been through and the feedback we've received, it has given us some really good. . . [pause] where you think aha that's why! Um, so some, maybe for some people, a bit basic information, but when you're a first-time parent, it's like "aha!"" (P4) "Yeah, understand what it is he wants. Um, try to put myself a bit in his place and like that, when he climbs up on me and says I should drive, what is it he wants then? And he calls for my husband 'ah okay, so we're going to play "catch" and drive around like crazy', right? So, those kinds of things." (P10) |
| 2. | Meeting the child's needs in comfort and in play | Theme 2 covers the parents' experienced changes in parent-child interactions. Primarily changes in the parents' behavior when accommodating the children's needs for comfort, closeness and especially exploration, but also encompassing parents' experiences of changes in their children's behavior, with some children seeking more emotional support and proximity as well as initiating playful interactions with the parent more often. | " [. . .] playing is much more than just sitting and saying 'well, then we said that,' it can also be *a presence*. And I've become much more aware of, um, that thing about following her and her initiatives and not just overriding the play, because then I feel like something else." (P1) ". . . because it gave some peace of mind about other things where one might have been a bit frustrated [. . .] and just thought 'now you're damn well going to bed because you're damn tired' instead of just saying 'let's spend five minutes giving a hug, brushing teeth in a calm manner, yes you need a glass of water afterwards or a glass of milk or something'. And then just take it slow. Where one might have been more confrontational before, right?" (P4) |
| 3. | Learning to prevent and manage conflicts is essential–diverse experiences of gains and progress | Theme 3 covers the parents' experiences of change in relation to conflict prevention and management. The theme reflects how parents experience behaving, feeling and thinking differently about conflictual encounters with their children, as well as they report of having acquired new skills or tools to prevent and manage conflicts. The theme also reflects the parents'´ experience that the change in this area was not sufficiently big, especially in terms having acquired sufficient skills to manage situations with a high level of conflict. | " [. . .] where before, I might have had, um, I wouldn't say more of a temper, but where I could quickly get worked up and become angry because he gets very hysterical when he doesn't get his way with something, you know. But now I can sort of step in and have that calmness, and say 'those are his feelings', 'that's how he feels' and articulate and acknowledge that [. . .] it's okay to get frustrated when you're told no, but that's just how it is. " (P6) " [. . .] but it's precisely when we get all the way out there and stand right on the edge and about to fall down, that you don't know what the heck to do. And there we are quite a few times, I think, during the week. Um, so— that's what I wish there had been [in the intervention, ed.], where you could see us [. . .] So maybe I had somehow wished, that we had come all the way out there where I just couldn't get the child to listen. Because that's where we need help." (P4) |
| 4. | "I'm actually not a bad parent"–new positive perspectives | Theme 4 covers the parents' experience of change in confidence and trust in their ability to be a good-enough parent prior to and post VIPP-SD. The theme also captures a transition in how the parents perceive the parent-child relationship, how they experience it to have developed and how they note positive changes in their feelings towards their child. | "[. . .] I have had an idea that, not that he didn't like me [his child, ed], but that we had difficulty interacting, but there I've seen that we absolutely don't, and that it's almost completely the opposite. And I think that's been really rewarding for me, because I've gotten such a huge positive confirmation that what we're doing is good, and what I'm doing with him is really good. So it's been really nice to find out" (P5) |

One parent described how his enhanced understanding of his child's emotional states helped him to change his once-negative interpretations of the child's behavior during conflicts, which also reduced his negative thoughts regarding the quality of their relationship. He described how he previously tended to think that the child's anger during conflicts reflected a dislike of him, whereas he was now able to understand the child's behavior as specific to the situation.

Other parents described similar insights, such as "of course the child gets angry, but the child is not angry with me, the child is angry with the situation" (P2). Parents also described how the VIPP-SD intervention had increased their knowledge of typical child development and behavior. This supported their improved understanding and acceptance of their own child's behaviors and signals and was often accompanied by feelings of relief following the acknowledgment of the child's behavior as typical or age appropriate. Some parents' descriptions further reflected how some experienced greater acceptance of their child's distinct qualities and individuality through their improved understanding.

## Theme 2: Meeting the child's needs in comfort and in play

Theme 2 centers on the parents' experiences of change in interactions, particularly how they responded to their children's needs differently following the VIPP-SD intervention. The parents' statements indicate that the improved understanding of child behavior and development also led the parents to *do* something differently: "So yeah, I think there are many things where we've luckily become wiser, and now can see what her needs are in a different way, and then also fulfill them in a different way." (P1)

Changes in the parents' behavior were exemplified by their descriptions of concrete situations. For example, instances of outbursts and frustration around bedtime were now handled with increased patience, psychical proximity, and affection. One parent's narrative illustrates how the new interpretations of child behavior led to changed parental responses characterized by warmth, comfort, and understanding:

"We also talked a lot about, uh, how sometimes she wants to be carried. And she wants to be carried, where it's like, "you're five, so you can walk by yourself," right? But, where we also agreed, well, she might also uh, she might be lacking that close contact in some way, because she's rejected it for so long. So maybe it's okay to pick her up and give her that, because being carried is not just a "I don't feel like walking by myself," there's so much more to it. There's presence, there's caressing, there's love, there's [. . .], it's so much more than just dragging your child because they don't want to walk. [. . .] So, we are much more receptive when she wants that affection and care, because she hasn't sought it in the same way before. Uh, so that has been really nice." (P1)

Parents, in general, reported having gained a new awareness of how frequently their children seek contact and emotional support, as well as an increased awareness of the significance of providing opportunities for physical proximity. For instance, prioritizing a quiet time on the sofa after returning home from childcare, engaging in conversation and reading, rather than immediately attending to practical tasks such as preparing dinner.

Parents shared experiences of change, both regarding their own behavior and that of their child, especially in the context of playing together. They repeatedly described how their own participation in play had changed, with the parents trying to follow the child's initiatives and letting the child explore based on his/her own needs. Some emphasized how a parent's role in play can be characterized by being present and supportive. The parents' efforts to take a step back and allow the child to lead during playtime were explicitly described as a particularly significant change in parental behavior:

"[. . .] I could see from the first films we made, that I might have been a bit too quick to, not intervene, but to step in and control the play. Because I could see that now my child might get a bit stuck, but there I have learned that he needs a bit more time to figure it out. Where

we as adults we're a bit like 'we don't want to sit and watch, let's just move on, let's find something else to do,' now I have become more aware that my child, or both my children, fundamentally need to be allowed to explore more [. . .]. It might take them two minutes to figure out what that car can do, or what that thing can do, but that's part of their learning process. Um, so I have definitely learned that about myself. That I need to step back, so it's not me who's leading the way." (P5)

As evident in this statement, the emphasis on allowing the child to explore at their own pace during play also highlights the parents' abilities and willingness to prioritize the needs of the children over their own agendas.

The shift in how the parents engaged with and followed their child's signals seemed to allow for a more playful attitude. Some parents' narratives included descriptions of how they increasingly supported pretend play and allowed themselves to be drawn into the world of imagination. Additionally, some reported being more aware of how important joyful playing with their parent is for the child, as well as how children initiate sharing joyful emotions with the parent:

"That made me aware of how much my child actually seeks my recognition through his eye contact with me when he is happy about something. Because the first thing he does is to look up at me and be acknowledged with a glance, a smile, or something else. So, I think about that much more now than I did before." (P8)

Apart from a shift in *the way* the parents engage in play, the parents also expressed new insights into the importance and positive outcomes of playing together, leading to a renewed focus on prioritizing playing and having fun together. Finally, some parents reported a noticeable *increase* in their children's initiatives regarding interactive activities following the intervention, and some described how their child now prefers playing together instead of alone.

### Theme 3: Learning to prevent and manage conflicts is essential–diverse experiences of gains and progress

Theme 3 centers on the parents' experiences of change in relation to the prevention and management of conflict, which they expressed as a particularly important area for change. Most parents experienced improvements in their ability to set limits and deal with difficult situations in successful ways, but many also noted how they would have liked to receive even more guidance within the area of conflict management and how they wished more time had been spent focusing on conflicts. As such, Theme 3 captures the parents' diverse experiences within this area.

Most parents described how they still had conflicts with their child, but through VIPP-SD, they had learned new ways to handle them. This changed both the way they acted in difficult situations with their child and their subsequent thoughts and feelings. Prior to the intervention, some parents described feeling powerless during conflicts with their child, being anxious and uncertain about what to do:

"I'm also sitting here thinking that before, uh conflicts were something that could, well, [. . .] make a chill run down my spine, because 'oh, ohhh what now,' right. Whereas now, I think we've gained a newfound calmness, or at least I have, to resolve conflicts and trust that it's just a feeling and it will pass. So, um, what can one say, more energy to resolve conflicts maybe." (P1)

Parents described how conflicts could trigger them in ways where their own emotions would spiral, similar to the child's reaction, making conflicts even more heated. After the VIPP-SD intervention, parents described how they were now able "stay calm" (P6) while handling challenging situations with their child. This ability to better regulate their own emotions during conflicts, was reported by several parents and perceived as a significant game changer. Staying calm and not raising their voice at their child was experienced as preventing conflicts from escalating and resulted in shorter durations of conflicts with lower intensity. The increased ability to regulate their own emotions thereby also reflected a decrease in parenting behavior that could be characterized as more negative:

"Um, I've also been prone to raising my voice, because [...] when you're busy, sometimes you raise your voice because as a parent, you think they'll listen more. And it just doesn't happen. So, it's also sometimes trying to keep your volume down, mean and say things the same way, but without raising your voice. Sometimes that in itself has made him actually shift down a gear or listen better." (P6)

Parents further reported gaining new skills through the VIPP-SD intervention to prevent and manage conflicts, so they now had a "box of tools" to guide their behavior (e.g., preparing the child for upcoming changes, only making one request at a time, distraction and positive reinforcement of appropriate behavior, using "the sensitive pause" when conflicts become particularly heated etc.).

In general, parents reported a more flexible approach to limit-setting and demands after receiving VIPP-SD. For example, some parents described how they increasingly took the child's mental state into account before asking the child to complete tasks or adhere to demands. Parents described becoming more patient with the child and that this eased conflicts and sometimes even prevented them from occurring. They explained how they now made a greater effort to understand what triggers the child's frustration, with some parents describing how they were now better able to empathize with their child during conflicts. The acknowledging stance was described by some as a significant change compared to how they used to handle conflicts prior to receiving VIPP-SD:

" [...] then I actually use that thing of trying to sit down in there completely calm, and simply say to him 'I can really see that you're upset,' 'I can see you got mad at me when I said no to the orange, but it's so you can eat your dinner, then you can have it afterwards.' Then he might be angry and upset, but I stay there and embrace it. And there can be days where I can be in there and embrace it, and some days where I can feel that I need to take a break, and then I come back again and talk with him about the feelings he has. Um, and that's something I haven't done much in that way before." (P6)

The major changes, according to the parents, were to be found within themselves, in their interactions with their child and their behavioral pattern, whereas the child, to some extent, behaved and responded as usual in relation to conflicts. However, some also described notable positive changes in the child's behavior. For example, some parents shared examples wherein the child's behavior during conflicts exhibited fewer externalizing tendencies than before; for instance, the child did not hit, bite or throw things around anymore, and some children were more compliant.

Although many parents reported that VIPP-SD supported conflict management, experiences varied regarding the extent of outcomes within this domain. Several expressed a need for more support in this area. Some parents felt that sessions with little or no focus on conflicts

were a wasted opportunity, as their primary concern was learning how to deal with the child's externalizing behavior. They considered conflicts the key issue, making subjects, such as play, seem irrelevant. When asked about unfulfilled expectations, some parents' accounts suggested that they still felt a need for tangible, practical knowledge after VIPP-SD, particularly concerning understanding and knowing how to handle "the real conflicts" and typical challenging situations: "Um, or if the intervener had been with us in the supermarket, she could have given advice on, well, how [. . .] to handle it, and how not to handle it. What would one say to the child." (P8) One parent further described changing his parenting approach in some situations, though he did not feel the intervention had improved his conflict-management skills specifically.

### Theme 4: "I'm actually not a bad parent"–new positive perspectives

A common pattern centers on how the parents experienced a boost in their parental confidence through participating in VIPP-SD. While some parents entered the project with self-blame and a focus on the challenges in everyday life, they described how they became aware of the things they did well and their successes in parenting. Realizing all the things that worked made a significant impression on many parents and left them feeling more competent and capable as parents:

"Because I feel like before, I was a bit worried and very much in that negative spiral. You only saw the challenges and the issues [. . .]. And I feel like VIPP has shed light on the good things and what you actually do without even thinking about it. So that [. . .] has made me think 'okay, I'm actually not as bad a parent as I felt I was,' because I actually do some things already that I just hadn't thought about myself. And that was actually really nice, because I've had such a [. . .] guilty conscience. [. . .], so it's been really nice to have it illuminated from the outside in a different way." (P6)

Several parents described how the intervention led them to recognize their importance to their child, fostering an understanding that their child liked them and wanted to spend time with them, a realization they had not been certain of prior to VIPP-SD. Through the intervention, parents came to recognize that even though the catalyst for their participation in the intervention project was embedded in the conflicts and everyday struggles with their child, the interaction between the parent and child was full of positive moments. For many parents this realization came as a surprise, and they described how it made them reconnect with feelings of love and warmth towards their child:

"We've become a bit more in symbiosis in some way. And I also said to the intervener, every time she was out here, when we were going to watch what she had filmed, I could feel in some way, that I just loved my child a bit more and wanted to bring her home, because [. . .] you don't often see yourself from the outside, but it was just [. . .] so clear, how much love there was between us when I watched these video clips. So, it was really nice." (P1)

## Discussion

This qualitative study explored parents' experiences of change following the VIPP-SD intervention. Our analysis resulted in four themes. The themes are highly interrelated, influencing each other in multiple ways.

Theme 1 related to the parents' experiencing an improved ability to interpret their children's signals through an enhanced understanding of how children's behavior reflects their internal states and developmental stage. This finding could be interpreted as an improvement in the parents' reflective functioning, denoting their capacity to understand and interpret the child in relation to mental states such as feelings and desires [45]. This capacity is closely linked to Ainsworth's definition of parental sensitivity, where sensitive behavior presupposes that the parent is able to correctly interpret the child's signals [28]. In line with this, a number of studies on reflective functioning and parental mentalization have documented the importance of the parents' capacity to accurately interpret children's internal states for parental sensitivity, child attachment, socio-emotional development, and reduced externalizing behavior problems [45–50]. The stimulation of parents' reflective functioning is suggested to be one of the pathways through which VIPP-SD enhances sensitivity [19]. The parents valued this outcome and experienced it as helpful and relevant. This finding furthermore supports previous findings from qualitative studies on parental experiences of outcomes of VIPP-SD/adaptations of VIPP-SD [32–37].

The parents reported behavioral changes towards more sensitive responsiveness, noting their increased ability to respond more adequately to their children's needs. These changes thus correspond to the last component in the definition of parental sensitivity (i.e. adequately responding to the child's signals) [28], as well as the primary aim of VIPP-SD [20], and are also in line with the findings from previous qualitative studies [32–37]. The changes in parental responding described by the parents reflect an increased effort to attune to the child's feelings and emotional needs, both in playful situations and in difficult situations. The parents described their efforts to meet the child's needs for comfort and emotional regulation, along with supporting child exploration in new ways, such as allowing the child to take the lead during play. The last point was especially evident and may signal a possible decrease in intrusive parenting behaviors. The parents emphasized the significance of joint play interactions and reported changes in their parenting attitudes and behavior, especially within the context of play. This finding is in line with results from Oliveira and colleagues [35], who also found new perspectives on the importance of play among caregivers as well as on how caregivers engaged in play following participation in their adapted version of VIPP-SD. This finding may have significance for child externalizing behavior problems, as studies report that parent-child joint play is related to reductions in conduct behavioral problems [51]. For instance, longer durations of joint play at age 3 predict improvements in conduct problems at age 4, independent of other risk factors [51]. Our analysis also suggested an increased awareness of moments of shared joy, as well as an increase in parental playfulness, exemplified by engaging in activities such as imaginary play. Embedded within VIPP-SD is a specific emphasis on supporting shared emotions and illustrating the significance of positive emotional involvement in interactive play to the parent. Experiencing positive emotions is in general known to have many beneficial effects [52,53], and shared positive affect, in combination with adequate parenting behaviors, is specifically found to have positive effects on self-regulation and healthy socio-emotional development of the child [54–57]. From an attachment-theoretical perspective, the described changes in interactions can be seen as parents trying to be both the *secure base* for the child to explore from, and the *safe haven* for the child to return to, when the attachment system is active/when experiencing negative emotions [58]. Supporting sensitivity and secure attachment relationships are, as recounted earlier, the overarching aim of VIPP-SD, as both constructs are known to be crucial for child development, and for preventing/decreasing externalizing problems [13,17,58–62].

Theme 3 centered on conflict management and prevention; while some parents experienced positive changes in this area, others expressed an absence of "enough" change. In

particular, parents reported an improved ability to regulate their own emotions, to "stay calm" during conflicts, as an important change and outcome of the VIPP-SD. This is in line with other qualitative studies on parents' experiences with parenting interventions aiming at reducing externalizing behaviors, such as the Triple P-intervention and the Parent Management Training—the Oregon Model (PMTO™) [63,64]. Emotion regulation capacities are recognized as crucial in the context of parenting, where challenges and conflicts are inherent. Parents who possess limited emotion regulation skills are, consequently, at an increased risk of reacting to challenging situations with displays of more negative emotions, such as anger and hostility [65]. The enhanced understanding and reflection outlined in theme 1 were also perceived by the parents as helpful in terms of preventing and managing conflicts, as it increased their understanding and insights into the underlying reasons behind their child's challenging behavior. This, in turn, appeared to support parental emotion regulation and facilitated parents' feelings of empathy and warmth towards their child during conflicts, as well as acknowledgement and verbal validation of the child's difficult emotions.

Parents in our study reported learning distinct helpful disciplining strategies, which they tried to incorporate in their daily lives. These were primarily used preventively, to avoid conflicts happening, such as positive reinforcement, clear communication about demands and adapting to the child's pace and state, while the strategy "sensitive pause" was described as helpful during unfolding and heated conflicts. Acquiring hands-on strategies seemed to enhance parents' experience of agency during challenging encounters with their child. Our findings correspond to results from previous qualitative investigations of adaptations of VIPP-SD, which suggested that providing parents/caregivers with a variety of specific skills to prevent and manage difficult behaviors, was experienced as helpful [32,33,35], though one study also highlighted an experienced need for more guidance [33]. The parents in our study did, likewise, express a wish for even more direct guidance, especially regarding what they called "the *real* conflicts" because, as one parent described, "it's precisely when we get all the way out there and stand right on the edge about to fall down, that you don't know what to do" (P13). This indicates that they still lacked sufficient knowledge on how to act in the peak of the conflict, or that they had trouble transferring learned skills and strategies to those situations.

The findings in theme 3 taken together suggest however, that the parents found elements inspired by both attachment theory and coercion theory helpful when trying to prevent and handle child externalizing behaviors, and that both approaches contribute to driving the experienced changes. This supports the idea that a framework for understanding the origins of externalizing problems and how to address them efficiently in interventions can benefit from bridging attachment theory and social learning theory [31,66]. Although several parents appear to learn skills to break and avoid coercive cycles and negative disciplining strategies, the diverse experiences and insufficient change illustrated in theme 3 also imply that the sensitive discipline component of VIPP-SD may not completely achieve its intended purpose in the current study.

It is further worth noting that the parents primarily reported changes within themselves, and less in the child's behavior in relation to conflicts. This is in line with the evidence-base for VIPP-SD, which points to VIPP-SD being especially effective in enhancing positive parenting behaviors and attitudes towards sensitivity and sensitive discipline, as well as improving child attachment [20]. This, combined with our finding that many parents requested more support with conflict management, indicates that further investigation is warranted to explore the most effective ways of intervening in families experiencing externalizing problems. This last perception is important, as an absence of expected change in areas deemed especially crucial for the parents may compromise the perceived meaningfulness of the intervention. Previous qualitative studies of adaptations of VIPP-SD, which also focused on parents with children exhibiting externalizing behaviors (targeting younger age-groups than the present study), found

similar experiences among the parents [33,36], emphasizing the potential for further development of the intervention when addressing externalizing behaviors. It is, however, important to consider that the intervention in the present study was offered as a preventive intervention for preschool children. Results from O´Farrelly and colleagues [21] indicate that the impact of the intervention on child externalizing behaviors might be more pronounced in populations with children exhibiting higher levels of symptoms, and a recent meta-analysis points to the intervention being more effective in this domain in studies with younger children [20].

Finally, parents experienced positive changes in their perception of their own parenting capabilities and in their relationship with their child. They experienced an increase in confidence and gained more trust in their ability to cope with conflicts and be good-enough parents, which is in line with previous findings [33,36]. While enhancing parental confidence is not the end goal of VIPP-SD, our analysis indicates that from the parents' point of view, feeling more confident in their parenting is a significant experienced change. In line with this, a review of studies that investigated parents' perceptions of parenting programs, found that parents consider an increase in experienced competence to handle difficult child behavior one of the most valuable outcomes of such interventions [67]. Increasing confidence might be important, especially in families where the child exhibits behavioral problems, as lower levels of parental confidence and higher levels of externalizing behavior in children have been previously linked [68]. Also, a growing body of research suggest that parenting programs can improve parental mental health and psychosocial functioning [69], which must be viewed as a valuable outcome in its own right. Our study indicates that the parents experience meaningful changes following VIPP-SD that not only holds the potential to support positive child development, which is the overarching goal of the intervention, but also reflect positive changes in relation to their own parenting journey.

## Implications for research

Even though a recent comprehensive study of the efficacy of VIPP-SD on externalizing behaviors show positive and promising results [21], participants in our and similar studies express a wish for more help in relation to managing conflicts and challenging behavior. This, combined with the absence of meta-analytical evidence for the effect of VIPP-SD on children's externalizing behaviors, underscores the need for future research to investigate whether the intervention could be even more efficient in addressing management of situations characterized by a high level of conflict and negative affect. This is important not only in terms of promoting efficacy but also in terms of making sure the intervention is experienced as meaningful to the participants, thereby preventing low engagement and attrition. Future studies should investigate which intervention components are not only most effective but also perceived as vital for desired change by parents of children with externalizing behavior, as well as potential inhibiting factors. This is important due to the flexibility of the method, which, on one hand, allows the intervention to be individualized, but on the other hand risks some learning messages being underrepresented in some VIPP-SD trajectories. This highlights the importance of future explorations of the key facilitators of change in VIPP-SD, to be able to ensure they are sufficiently delivered in the individual trajectories. Future studies could, moreover, include measures of reflective functioning/parental mentalization, parental emotion regulation capacities and parental self-efficacy as potential mediators or pathways for change.

## Strengths and limitations

The primary objective of the present study was to elicit a detailed analysis of participants' experiences of change following VIPP-SD. Qualitative research does not aim for generalizability

but strives for depth and nuanced insight, combined with taking measures to allow readers to be able to assess the degree of transferability to other contexts. We promoted credibility and transferability by employing strategies to enhance reflexivity, and by continuously seeking feedback from the research team to critically evaluate the research process and analysis, in addition to providing descriptions of the process of analysis, of the research setting, the context and participants, as well as the data collection methods.

The primary limitation of the study might be the homogenous sample. Firstly, our sample consisted of volunteer participants who completed the entire intervention. Though they did point to areas for improvements, they might have been more prone to elaborate on positive experiences than participants who dropped out. As the study focused on experienced change from participation, a prerequisite was completion of the intervention. Since the embedding feasibility study had a high proportion on non-completers, it would have been interesting to investigate more divergent experiences. Furthermore, our sample was resourceful in terms of education, marital status, and employment status. Additionally, our study included both mothers and fathers, but nine out of 11 participants were mothers. Research points to a need to explore the experience of fathers specifically, who are underrepresented in studies of parenting interventions [70,71].

While most participants reported predominantly positive experiences with the intervention, variations in experiences were observed in both the degree and domains of change. Our investigation did not explore factors such as individual differences among participants, which could be a pivotal area for future research exploring 'what works for whom'. Furthermore, we were unable to investigate the potential long-term effects of the intervention. For instance, some parents might observe additional positive changes in their children's behavior over an extended period if the changes in parental behavior are maintained. Conversely, we cannot ascertain whether the reported changes endure over time.

Our focus on the most prominent patterns of change relating specifically to the parent and child involved in the intervention, resulted in some reported changes not being covered in the present analysis, such as improved parental teamwork and broader family dynamics.

Finally, the study was conducted during 2020–2023 while COVID-19 still affected the Danish society. It is unknown if and how this affected the experiences with participation. None of the participants in the sample mentioned the pandemic when asked about important factors affecting outcomes, but the length of many intervention trajectories did exceed the recommended timeframe because of repeated cancellations and re-bookings that might have been related to the effects of the pandemic.

## Conclusion

This qualitative study was conducted to explore parental experiences of change following VIPP-SD in families with a child exhibiting externalizing behaviors. The parents' experienced changes reported in the present study overall indicate that VIPP-SD attains its goal of supporting different prerequisites and vital aspects of sensitive parenting, and secure attachment relationships. While this perhaps was to be expected following the quantitative evidence supporting the same [20], the result is nonetheless encouraging, as the change towards (perhaps even more) sensitive parenting behaviors is something that the parents themselves were aware of (though expressed in their own words) and in general experienced as positive, relevant and helpful changes. The results are, and should be seen as, a contribution to understanding the experience of change. In this case, experiences largely support the hypothesized outcomes and mechanisms of change behind VIPP-SD, especially regarding improving parental sensitivity. However, they also highlight areas related to the sensitive discipline component

of VIPP-SD, where parents do not perceive the intervention as effective enough or meeting expectations, particularly regarding the capacity to handle children's externalizing behaviors.

## Acknowledgments

We thank all the families who participated in the current study. We are very grateful for their contribution to our research into parental experiences of change following VIPP-SD.

## Author Contributions

**Conceptualization:** Nina M. Lassen, Tine Steenhoff, Mette S. Væver.

**Formal analysis:** Nina M. Lassen, Tine Steenhoff.

**Investigation:** Nina M. Lassen, Tine Steenhoff.

**Project administration:** Nina M. Lassen, Tine Steenhoff.

**Supervision:** Bryan Cleal, Mette S. Væver.

**Writing – original draft:** Nina M. Lassen.

**Writing – review & editing:** Tine Steenhoff, Bryan Cleal, Amanda Frees, Mette S. Væver.

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
