## [Decision Letter · Decision Letter 0]

14 Aug 2024

PONE-D-24-23724Parental experience of change following VIPP-SD in families with preschool children with externalizing behavior:

A qualitative interview studyPLOS ONE

Dear Dr. Lassen,

Thank you for submitting your manuscript to PLOS ONE. After careful consideration, we feel that it has merit but does not fully meet PLOS ONE’s publication criteria as it currently stands. Therefore, we invite you to submit a revised version of the manuscript that addresses the points raised during the review process.

Thank you for submitting your manuscript to PLOS ONE. I appreciate the opportunity to review your work, which is well-written and addresses an important aspect of interventions—specifically, the parents' experiences. The methodology is clearly articulated, and the article flows well, making it an important contribution to the field.

I concur with Reviewer 1's observation that the paper is somewhat lengthy and could benefit from being shortened for greater impact and clarity. Here are a few additional comments and suggestions:

**Line 227 (Intervention):** Could you clarify who delivered the VIPP-SD intervention? Was the VIPP-SD-trained childcare provider employed at the child’s daycare center, and was the intervention conducted at the daycare or in the child's home?**Line 267:** I noticed that you mentioned using Nvivo for coding in Line 318. It would be helpful to combine this information earlier on to avoid any confusion about the methods used.**Line 501:** Could you provide more specific data on how many parents reported observing less externalizing behavior in their children? Adding numbers where possible would strengthen the connection between your data and the themes you present.**Table 3 Integration:** Table 3 is not well-integrated into the text. I recommend omitting Table 2 (example) and starting the analysis section by presenting the themes in Table 3. This will help to better incorporate Table 3 into the analysis and improve the overall flow.

Thank you again for your submission. I look forward to your revisions and believe they will further enhance the quality of this important work.

We look forward to receiving your revised manuscript.

Kind regards,

Maiken Pontoppidan

Academic Editor

PLOS ONE

Journal Requirements:

2. You have indicated that data is available from "jan.nielsen@psy.ku.dk". 

Please can we ask you to provide us with a general contact email address for the data requests, so readers can request access in perpetuity. If a general email is not available please provide a link to a website where readers can obtain access to data. 

Reviewers' comments:

Reviewer's Responses to Questions

**Comments to the Author**

1. Is the manuscript technically sound, and do the data support the conclusions?

Reviewer #1: Yes

Reviewer #2: Yes

2. Has the statistical analysis been performed appropriately and rigorously? 

Reviewer #1: N/A

Reviewer #2: Yes

3. Have the authors made all data underlying the findings in their manuscript fully available?

Reviewer #1: No

Reviewer #2: Yes

4. Is the manuscript presented in an intelligible fashion and written in standard English?

Reviewer #1: Yes

Reviewer #2: Yes

5. Review Comments to the Author

Reviewer #1: Thank you for the opportunity to read about this work to understand the mechanisms of how VIPP-SD worked for this group of parents. Particular strengths of the manuscript include:

The data analysis and reflexivity section is thoughtful, relevant and shows a good understanding of the use of reflexive TA

The detail on the intervention itself is comprehensive and transparent which helps enormously in qualitative studies where the detail of the intervention is needed to understand how and why it might function.

The use of semantic and latent coding has led to well-developed, interesting interpretations of the data, well evidenced, with coherent themes forming plausible central organising concepts. Bravo.

The discussion does an excellent job of situating the findings into the context of other research into parenting interventions and pays particular attention to those interventions that propose similar mechanisms.

Minor suggestions to improve the manuscript include:

The participants section should move to the beginning of the Analysis (which could be more usefully renamed 'findings' or more traditionally 'results')

I'm interested in outcomes for parents as a meaningful goal in and of themselves - the discussion hints at this but if the authors agree, it could be made clearer that improving parents' experiences is valid whether or not it leads to improved outcomes for the child. Future research that values how parents experience their parenting journey and the outcomes that are important to them may be valuable.

The manuscript mentions the eligibility criteria as relating to child behaviour but not to parenting behaviour, was this collected pre and post training? Do we know which parenting behaviours changed in the group? Were any of them exhibiting harsh and insensitive parenting prior to taking part? Including this (just as a description of the parents) would help contextualise the findings.

The manuscript reads very much as a chapter in a thesis, and while it is an excellent chapter, at over 10,000 words I wonder if it will lose attention from the intended audience by being too long as a published paper. I think some careful editing to bring the word count down without losing the important details would make it easier to engage with from a reader's perspective.

There's a minor typo on p32 "qualitative research does not" (not "do")

Best wishes,

Dr Anna Pease

Reviewer #2: The research addresses an important aspect of management (parental experiences). It is relevant to readership and contributes to the understanding of issues within the field.

The manuscript is well-written and the methodology is clear. The flow of the article is easy to follow and readable.

6. PLOS authors have the option to publish the peer review history of their article (what does this mean?). If published, this will include your full peer review and any attached files.

Reviewer #1: **Yes: **Anna Pease

Reviewer #2: No

---

## [Author Response · Author response to Decision Letter 0]

2 Sep 2024

Dear Maiken Pontoppidan

We thank you and the reviewers for inviting us to revise and resubmit our manuscript. 

We have carefully considered and revised our manuscript taking into account the valuable feedback provided by you and the reviewers.

Below we address the comments and suggestions point by point with reference to the location of the changes in the revised manuscript.

Please note that in the new version of the manuscript we have made additional changes that were not due to the reviewer comments:

1. It came to our attention that we had stated that the interveners participated in a five-day training program, but it was only four days. This has now been corrected (line 213).

2. We have added two illustrative quotes (one from the omitted table (2)) into existing table 2 (previously table 3). 

3. We have occasionally made minor corrections as small mistakes or typos came to our attention while shortening the paper (e.g. changed the wording “parents” to “caregivers” in line 549-551, since the mentioned study was regarding foster carers experience) 

On the behalf of my co-authors, 

Kind regards, 

Nina Marie Lassen

 Editors comments: 

Thank you for submitting your manuscript to PLOS ONE. I appreciate the opportunity to review your work, which is well-written and addresses an important aspect of interventions—specifically, the parents' experiences. The methodology is clearly articulated, and the article flows well, making it an important contribution to the field.

Thank you for the positive feedback and for acknowledging the importance of investigating parental experiences when evaluating parenting interventions. 

I concur with Reviewer 1's observation that the paper is somewhat lengthy and could benefit from being shortened for greater impact and clarity. 

ANSWER: We agree that the paper benefits from being shorter than initially proposed. We have now revised the entire paper and shortened it wherever possible. Several minor changes have been made, along with some larger revisions. All corrections are visible in the document file with track-changes. 

Here are a few additional comments and suggestions:

• Line 227 (Intervention): Could you clarify who delivered the VIPP-SD intervention? Was the VIPP-SD-trained childcare provider employed at the child’s daycare center, and was the intervention conducted at the daycare or in the child's home?

ANSWER: We have now clarified that the intervention took place in the family’s home (line 210). Additionally, we have added a line specifying who delivered the intervention (line 211-213). It was delivered by either childcare providers, who did not work in the family’s childcare center, but in another childcare center in the municipality, or by childcare providers who were affiliated as educational/pedagogical consultants within the municipality. The latter group were not regularly present in the child’s childcare center and did not perform typical pedagogical care tasks. 

• Line 267: I noticed that you mentioned using Nvivo for coding in Line 318. It would be helpful to combine this information earlier on to avoid any confusion about the methods used. 

ANSWER: We now inform of the use in Nvivo earlier on, particularly in Line 255.

• Line 501: Could you provide more specific data on how many parents reported observing less externalizing behavior in their children? Adding numbers where possible would strengthen the connection between your data and the themes you present.

ANSWER: Thank you for this comment. We understand why it would be interesting to know how many parents reported such changes. However, in line with the Reflexive Thematic Analysis method, we have deliberately sought not to quantify the data in relation to the themes because this would, in our reading of the approach, be at odds with its underlying methodological principles. Reflexive TA explicitly eschews approaches that seek to link quantity of the data to the quality of the theme. The theme is generated on the basis of its analytical relevance to the central organizing concepts of the analysis. Of course, it is true to say the volume of data on specific topics will influence analytical choices and to some extent this is evinced in our narrative, where we frequently allude to vague quantifications, such as the ‘majority of’, ‘many’ or ‘some’ participants. While we recognize that not adding numbers might be frustrating for some readers, we are hesitant to provide a more quantified rendering of the data because it risks the inference that the quality of the theme (and the relevance of the different experiences) is determined by the number of times it is mentioned or by the number of individuals who mention it. Put differently, from the analytical perspective of Reflexive TA, whether all, some or only one person mentioned a particular topic does not, in and of itself, determine how we should value the theme in a broader analytical purview. 

• Table 3 Integration: Table 3 is not well-integrated into the text. I recommend omitting Table 2 (example) and starting the analysis section by presenting the themes in Table 3. This will help to better incorporate Table 3 into the analysis and improve the overall flow.

ANSWER: Thank you for the recommendations. We agree that it is more appropriate to present the table 3 (now renamed table 2) in the beginning of the analysis (now renamed “Results” following reviewer 1’s requests). It has been moved and we have omitted table 2. 

Review Comments to the Author

Reviewer #1: 

Thank you for the opportunity to read about this work to understand the mechanisms of how VIPP-SD worked for this group of parents. Particular strengths of the manuscript include:

The data analysis and reflexivity section is thoughtful, relevant and shows a good understanding of the use of reflexive TA

The detail on the intervention itself is comprehensive and transparent which helps enormously in qualitative studies where the detail of the intervention is needed to understand how and why it might function.

The use of semantic and latent coding has led to well-developed, interesting interpretations of the data, well evidenced, with coherent themes forming plausible central organising concepts. Bravo.

The discussion does an excellent job of situating the findings into the context of other research into parenting interventions and pays particular attention to those interventions that propose similar mechanisms.

Thank you very much for your positive and detailed feedback, it is very helpful. 

Minor suggestions to improve the manuscript include:

The participants section should move to the beginning of the Analysis (which could be more usefully renamed 'findings' or more traditionally 'results')

ANSWER: Thank you for these suggestions. We have now renamed the previous “analysis”-section, so it’s called “results” instead. 

In terms of the participant section we are hesitant to move it to the results section as suggested. It is, to the best of our knowledge, common to provide details on participants in the method section, also in published papers utilizing qualitative methods such as Thematic Analysis. 

I'm interested in outcomes for parents as a meaningful goal in and of themselves - the discussion hints at this but if the authors agree, it could be made clearer that improving parents' experiences is valid whether or not it leads to improved outcomes for the child. Future research that values how parents experience their parenting journey and the outcomes that are important to them may be valuable.

ANSWER: Thank you for highlighting this important point. We agree that the positive outcomes in relation to outcomes relating to parents’ experiences of parenthood and parenting are valid and meaningful in their own right. We have now addressed this further in line 642-647.

The manuscript mentions the eligibility criteria as relating to child behaviour but not to parenting behaviour, was this collected pre and post training? Do we know which parenting behaviours changed in the group? Were any of them exhibiting harsh and insensitive parenting prior to taking part? Including this (just as a description of the parents) would help contextualise the findings.

ANSWER: Thank you for your comment. We understand that it would be informative to include such outcomes to contextualize the findings. In this paper our primary aim was to explore parents’ experiences qualitatively, and though including quantified measures would help to contextualize these experiences, we do believe the parents’ experiences provide important information in itself. We did use more (currently not evaluated) measures pre and post intervention (surveys and interactions). However, the sample is very small and the overall aim of the study was to explore the feasibility of VIPP-SD and we believe that an examination of the parents’ experience of the intervention is a very first important step in this. However, we hope to include some of the more quantitative data in future papers. 

The manuscript reads very much as a chapter in a thesis, and while it is an excellent chapter, at over 10,000 words I wonder if it will lose attention from the intended audience by being too long as a published paper. I think some careful editing to bring the word count down without losing the important details would make it easier to engage with from a reader's perspective.

ANSWER: We agree that the paper benefits from being shortened. As also replied in relation to Editors related comment above, we have now revised the entire paper and shortened it wherever possible (it is now very close to 10.000 words, including references, excluding tables and abstract). Several minor changes have been made, along with some larger revisions. All corrections are visible in the document file with track-changes.

There's a minor typo on p32 "qualitative research does not" (not "do")

ANSWER: Thank you for pointing it out, it has been corrected. 

Best wishes,

Dr Anna Pease

Reviewer #2: 

The research addresses an important aspect of management (parental experiences). It is relevant to readership and contributes to the understanding of issues within the field.

The manuscript is well-written and the methodology is clear. The flow of the article is easy to follow and readable.

Thank you for the positive feedback and for appreciating the study’s contribution to the field.

---

## [Decision Letter · Decision Letter 1]

10 Oct 2024

Parental experience of change following VIPP-SD in families with preschool children with externalizing behavior:

A qualitative interview study

PONE-D-24-23724R1

Dear Dr. Lassen,

We’re pleased to inform you that your manuscript has been judged scientifically suitable for publication and will be formally accepted for publication once it meets all outstanding technical requirements.

Kind regards,

Maiken Pontoppidan

Academic Editor

PLOS ONE

Additional Editor Comments (optional):

Thank you for resubmitting your manuscript titled "Parental Experience of Change Following VIPP-SD in Families with Preschool Children with Externalizing Behavior: A Qualitative Interview Study" to PLOS ONE.

I appreciate the opportunity to review your revised manuscript. The changes you have made further enhance the clarity and impact of your work. The manuscript continues to address an important aspect of behavioral interventions by highlighting the parents' experiences.

The revisions have strengthened the methodology and presentation of findings, making the paper a valuable contribution to the field. I do agree with Reviewer 1, however, that the paper remains a bit long. A more concise presentation of certain sections could further improve its readability without sacrificing the quality or depth of your analysis.

Thank you for your careful attention to the feedback provided during the review process. I look forward to seeing this important work published.

Reviewers' comments:

Reviewer's Responses to Questions

**Comments to the Author**

1. If the authors have adequately addressed your comments raised in a previous round of review and you feel that this manuscript is now acceptable for publication, you may indicate that here to bypass the “Comments to the Author” section, enter your conflict of interest statement in the “Confidential to Editor” section, and submit your "Accept" recommendation.

Reviewer #1: All comments have been addressed

2. Is the manuscript technically sound, and do the data support the conclusions?

Reviewer #1: Yes

3. Has the statistical analysis been performed appropriately and rigorously? 

Reviewer #1: N/A

4. Have the authors made all data underlying the findings in their manuscript fully available?

Reviewer #1: No

5. Is the manuscript presented in an intelligible fashion and written in standard English?

Reviewer #1: Yes

6. Review Comments to the Author

Reviewer #1: Thank you for your responses to the first review. The manuscript is still quite long but all the comments have been addressed well.

7. PLOS authors have the option to publish the peer review history of their article (what does this mean?). If published, this will include your full peer review and any attached files.

Reviewer #1: **Yes: **Anna Pease

---

## [Editor Report · Acceptance letter]

1 Nov 2024

PONE-D-24-23724R1 

PLOS ONE

Dear Dr. Lassen, 

I'm pleased to inform you that your manuscript has been deemed suitable for publication in PLOS ONE. Congratulations! Your manuscript is now being handed over to our production team.

Kind regards, 

on behalf of

Dr. Maiken Pontoppidan 

Academic Editor

PLOS ONE